# Network Analysis of the Social Environment Relative to Preference for and Tolerance of Exercise Intensity in CrossFit Gyms

**DOI:** 10.3390/ijerph17228370

**Published:** 2020-11-12

**Authors:** Megan S. Patterson, Katie M. Heinrich, Tyler Prochnow, Taylor Graves-Boswell, Mandy N. Spadine

**Affiliations:** 1College of Education and Human Development, Texas A&M University, College Station, TX 77845, USA; tboswell0628@tamu.edu (T.G.-B.); mandyspadine@tamu.edu (M.N.S.); 2College of Health and Human Sciences, Kansas State University, Manhattan, KS 66506, USA; kmhphd@ksu.edu; 3Robbins College of Health and Human Sciences, Baylor University, Waco, TX 76706, USA; tyler_prochnow1@baylor.edu

**Keywords:** social networks, social network analysis, high-intensity functional training, group exercise, sense of community

## Abstract

Known for its ability to improve fitness and health, high-intensity functional training (HIFT) focuses on functional movements completed at high intensities, often yielding outcomes superior to repetitive aerobic workouts. Preference for and tolerance of high-intensity exercise are associated with enjoyment of and adherence to HIFT. Similarly, the social environment present within CrossFit, a popular group-based HIFT modality, is important to the enjoyment of and adherence to HIFT. This study aimed to test whether preference and tolerance were related to social connections within CrossFit networks. Linear network autocorrelation models (LNAMs) and exponential random graph models (ERGMs) were computed on sociometric and attribute data from members of three CrossFit networks (n = 197). LNAMs showed the preference and tolerance scores of someone’s social connections were associated with their own in all three gyms, and ERGMs demonstrated preference and tolerance scores were associated with the presence of social ties within all networks. This study is the first to provide evidence for a relationship between social connections and preference and tolerance. Future longitudinal research is needed to determine if the social environment may influence and optimize a person’s preference of and tolerance for HIFT.

## 1. Introduction

Despite overwhelming evidence of the health benefits related to regular exercise participation [1,2], the majority of U.S. adults do not meet physical activity guidelines [3,4]. The World Health Organization [5] and US Department of Health and Human Services [6] recommends adults engage in aerobic activity for at least 150 min at moderate intensity or 75 min at vigorous intensity each week, in addition to incorporating resistance training on at least two days of every week. Many adults cite a lack of time and enjoyment as primary barriers to engaging in consistent and effective exercise routines [7]. A potential solution that provides fitness and health improvements in less time per week than current guidelines is a relatively new method called high-intensity functional training (HIFT) [8]. HIFT is a “training style [or program] that incorporates functional, multimodal movements, performed at relatively high intensity, and is designed to improve parameters of general physical fitness and performance” [9].

Studies comparing HIFT-style workouts to repetitive aerobic exercises (e.g., jogging, using an elliptical machine) found HIFT elicits greater lean muscle mass, thereby improving cardiovascular endurance, strength, and flexibility [10,11,12]. Further physical benefits of HIFT include improvements in maximal oxygen consumption [11,13], decreases in body fat [11,13], as well as improvements in bone mineral content [13]. In addition to physical health benefits, investigators have reported higher levels of enjoyment among group-based HIFT participants than those engaged in more traditional resistance training programs [8,14], which facilitates initiation and adherence to exercise [9,15]. Based on its associated health benefits compared to traditional aerobic exercise, coupled with a reduction in time required to achieve these benefits, HIFT could be an ideal exercise modality to promote among US adults.

Most studies investigating HIFT have involved CrossFit workouts [8,10,12,14], which use group-based HIFT to combine aerobic and resistance exercises that focus on functional (multi-joint) movements within a group-exercise environment. CrossFit members participate in a “workout of the day” (WOD) in a class-based format, where coaches explain the WOD and attend to participants throughout the workout, with opportunities to modify or scale exercises accordingly to fitness and skill level [16]. Previous research suggests group-exercise environments are often more effective in delivering health benefits, such as enhanced mental health [17,18], improved sense of community and belonging [15], and greater enjoyment and adherence to exercise [19,20] when compared to independent exercise. Studies show similar effects among CrossFit participants, with evidence that CrossFit can be a supportive social environment for its members, providing them social capital and support that facilitates physical and mental health benefits [15,21]. For example, a recent qualitative study on CrossFit members and coaches showed people who are initially intimidated by CrossFit-style workouts overcome their initial fears and develop greater enjoyment of HIFT over time, in large part due to the supportive social environment and ability to self-select their intensity during workouts [19].

According to previous research, a key factor related to someone’s affinity for, and ultimately enjoyment of, HIFT is a preference for and tolerance of high-intensity exercise [22,23]. According to Ekkekakis and Petruzzello [24], preference for exercise intensity is defined as a “predisposition to select a particular level of exercise intensity when given the opportunity (e.g., when engaging in self-selected or unsupervised exercise),” while tolerance of exercise intensity is defined as “a trait that influences one’s ability to continue exercising at an imposed level of intensity beyond the point at which the activity becomes uncomfortable or unpleasant” (p. 354). In other words, tolerance for exercise intensity relates to a type of resilience when engaging in HIFT, while preference is representative of an individual’s choice to stay above an intensity threshold [25].

Most research concerning the preference for and tolerance of high-intensity exercise has been conducted on independent exercisers [22,24,26,27]. In these studies, preference and tolerance are described as dispositions that go unchanged despite someone engaging in more exercise or experiencing fitness improvements. For example, Hall and colleagues studied 42 firefighters involved in a six-week training program. At the end of the program, despite reporting improved fitness scores (e.g., 1-min repetitions of pushups and sit-ups, 1.5 mile run time), preference and tolerance scores did not increase significantly among the firefighters, supporting the notion that preference and tolerance scores are a stable disposition [27]. However, there is some evidence that suggests self-selecting intensity levels during workouts can improve tolerance levels for exercisers [28,29,30], revealing the possibility that environmental factors (e.g., self-selecting intensity) may work to optimize a person’s preference and tolerance.

The association between the social environment and preference and tolerance remains unstudied, with the potential for preference and tolerance scores to drive social connections created within CrossFit networks, along with the possibility for social connections to influence someone’s preference and tolerance scores. Specifically, we aim to explore the possibility of social selection, social influence, and social context relative to preference and tolerance scores. Social selection eludes to the propensity for people to connect with others they are similar to on a given trait or characteristic [31,32,33], indicating a characteristic (e.g., preference and tolerance) ultimately prompts the social connection between two people, and therefore serves to drive social connections. Contrarily, social influence suggests people become more like their social ties over time, meaning the social connection prompts changes in the outcome or characteristic so that two people become more alike [33]. Finally, social context refers to the simultaneous development of similar characteristics due to the influence of the shared environment [32]. Given the evidence that the social environment of group-based HIFT programs yields similar outcomes for participants as preference and tolerance (i.e., enjoyment, adherence) [10,19,21], we wondered if the preference and tolerance scores of CrossFit participants might be associated with social selection, social influence, and/or social context.

Therefore, the purpose of this study was to assess the social environment of CrossFit relative to preference and tolerance scores among members. Specifically, we tested whether preference and tolerance scores were associated with the presence of social ties within CrossFit networks, as well as whether preference and tolerance scores of one’s social ties correlate with their own preference and tolerance scores. Thus, we answered two major research questions in this study:Do the preference and tolerance scores of a person’s social connections relate to their own preference and tolerance scores after controlling for other factors including demographic information, duration of time as a CrossFit member, weekly class attendance rates, depressive symptoms, personality, and sense of community variables (i.e., is there evidence of social influence on preference and tolerance scores among CrossFit members)?What individual and social factors, including preference and tolerance scores, are associated with social connections present within CrossFit networks?

## 2. Materials and Methods

### 2.1. Participants and Procedure

All members of three CrossFit gyms (two gyms in Texas, one gym in Kansas) were invited to participate in this study. Prior to data collection, researchers worked with gym owners to obtain a complete roster of members from each gym. Members at each respective gym were defined as paying individuals who participated in CrossFit classes (as opposed to personal training). Therefore, paying members of each CrossFit gym who participated in CrossFit classes were included in the study. Anyone under the age of 18, or those paying for alternative services at the gym (e.g., personal training, children’s programming) were excluded. Gym 1 consisted of 92 total members (50.0% male, M = 36.56 years, SD = 9.77), Gym 2 consisted of 40 total members (47.22% male, M = 43.31 years, SD = 15.73), and Gym 3 consisted of 158 total members (31.82%, M = 33.57 years, SD = 9.10), all of which received emails describing the study purpose with an invitation to participate in an online survey. After giving their electronic consent, respondents provided demographic, attribute (i.e., behavioral health data), and sociometric network data. All study procedures were approved by Texas A&M University’s institutional review board (which ethically is informed by the Declaration of Helsinki [34]) prior to data collection, and guidelines outlined in the Strengthening the Reporting of Observational Studies in Epidemiology (STROBE) were followed in this paper [35].

A sociometric network design was used to assess social relationships within the three CrossFit gyms, and how they relate to preference for and tolerance of exercise. Sociometric networks analysis (i.e., whole network analysis) requires the assessment of all members within a defined, bounded group [36,37]. Similar to most survey research, network analyses are subject to missing data [38,39]. Research suggests a 60% response rate from a given group (e.g., CrossFit gyms) when collecting sociometric network data allows for an adequate representation of the network to conduct analyses [40,41,42]. Each gym met this requirement. Gym 1 had 58 responses (63.0% response rate), Gym 2 had 31 responses (77.5% response rate), and Gym 3 had 108 responses (68.4% response rate), for a total of 197 respondents across the three gyms.

### 2.2. Measures

#### 2.2.1. Demographic/Background Information

Each member indicated their age (in years) at the time of survey completion, their gender (open response, coded into male = 0, female = 1, other = 2), race/ethnicity (1 = white, 2 = Black/African American, 3 = Hispanic, 4 = Asian, Pacific Islander, 5 = Indigenous Person, Native American, Hawaiian Native, 6 = Multiracial, 7 = Other, 8 = Prefer not to say), how long they had been a member of CrossFit (0 = less than 6 months, 1 = 6 months–1 year, 2 = 1–2 years, 3 = 2–3 years, 4 = 3–4 years, and 5 = 4+ years), and on average, how many CrossFit classes they attended per week (0 = less than once per week, 1 = once per week, 2 = twice per week, 3 = three times per week, 4 = four times per week, 5 = five times per week, 6 = more than five times per week).

#### 2.2.2. Preference and Tolerance

Preference and tolerance for exercise was assessed using the Preference for and Tolerance of the Intensity of Exercise Questionnaire (PRETIE-Q) developed by Ekkekakis, Hall, and Petruzzello [43]. The PRETIE-Q is a two-factor scale consisting of 16 items, eight of which measure preference and 8 measure tolerance. Each item is measured on a 5-point Likert scale (1 = I totally disagree, 2 = I disagree, 3 = Neutral, 4 = I agree, 5 = I totally agree), where respondents indicate their level of agreement to statements such as “I would rather work out at low-intensity levels for a long duration than at high-intensity levels for a short duration.” Preference and tolerance scores are created by reverse coding and then summing the appropriate items for each factor, with higher scores indicating a higher preference for and tolerance of high-intensity exercise. Previous studies demonstrated strong validity and reliability using the PRETIE-Q, with Cronbach’s alphas ranging from 0.81 to 0.90 [23,24,26] and test-retest reliability coefficients ranging from 0.67 to 0.90 [24,26].

Upon conducting initial reliability analyses on our data, we discovered the 16-items of the PRETIE-Q did not demonstrate appropriate internal consistency within our sample (Cronbach’s alpha < 0.50 for all three gyms). As a result, we conducted a confirmatory factor analysis to determine specific item loadings within our sample. The confirmatory factor analysis revealed five factors: the first consisted of four original “preference” PRETIE-Q items (e.g., “Exercising at a low intensity does not appeal to me at all”), the second consisted of four original “tolerance” PRETIE-Q items (e.g., “Feeling tired during exercise is my signal to slow down or stop”), and the final three factors consisted of a mix of the remaining eight preference and tolerance items from the PRETIE-Q. In order to keep the distinction between preference and tolerance, we decided to keep the first two factors, made up of four-items from the original preference and tolerance set of items. Subsequent reliability analyses using the eight items identified in the primary two factors from confirmatory factor analysis registered higher Cronbach’s alpha scores across all three gyms (Cronbach’s α = 0.76–0.78), and an overall sample alpha of 0.77. Based on these results, we created updated sum scores for preference and tolerance using only the eight items loading on the top two factors (four for preference and four for tolerance).

#### 2.2.3. Personality

The “big five” dimensions of personality (extraversion, agreeableness, conscientiousness, emotional stability, and openness to experiences) were measured using the Ten-Item Personality Inventory (TIPI) [44]. The TIPI’s validity has been assessed and recommended based on its utility for measuring the big five personality traits easily [45]. Each trait represents a spectrum (versus a dichotomy). For extraversion, lower scores indicate someone quieter and reserved, whereas higher scores represent someone who tends to be outgoing and warm. Lower conscientiousness scores represent someone who is more impulsive or disorganized, while higher scores indicate a dependable and hardworking individual. Agreeableness ranges from being critical or uncooperative on the lower end, to helpful and trusting on the higher end. Someone who scores low on emotional stability (also referred to as neuroticism), tends to be anxious, unhappy, or prone to negative emotions, while higher scores are indicative of someone who is more calm, even-tempered, and secure. Finally, openness to experiences ranges from practical, routined persons on the low end, to curious and spontaneous persons on the high end [46].

Each item on the TIPI measured the extent to which a pair of traits (e.g., anxious/easily upset; sympathetic/warm) applied to the respondent, with answer choices ranging from 1 to 7 (1 = Disagree Strongly, 7 = Agree Strongly). The study sample exhibited moderate internal consistency (Cronbach’s α = 0.60), congruent with the authors’ original intent to create a minimal item scale measuring broad items studies which often yield lower internal consistency coefficients [44,47,48]. Scores for each of the five personality traits were calculated by summing the two items associated with each factor.

#### 2.2.4. Depressive Symptoms

Depressive symptoms were derived from the eight-item version of the Personal Health Questionnaire (PHQ-8) [49], which measures symptoms of depression without a question of suicidality. The PHQ-8 uses a 4-point Likert scale (0 = not at all, 1 = several days, 2 = more than half the days, 3 = nearly every day), measuring an individual’s frequency of experiencing various symptoms. A total PHQ score was created by summing all items, with higher scores revealing more frequent and severe depressive symptoms. While true diagnoses should be verified by clinicians, scores of 10 or greater suggest evidence for major depression, and a score of 20 or above suggest evidence for severe major depression [50]. This scale has shown good internal (α = 0.89) and test-retest (Pearson’s r = 0.84) reliability in previous studies, as well as diagnostic validity (sensitivity: 88%; specificity: 88%) [49]. Our sample registered a Cronbach’s α of 0.86.

#### 2.2.5. Sense of Community

Sense of community (SOC) consists of environmental or community characteristics that lead to individuals feeling a sense of belonging and social support at the group-level [51]. SOC was measured in this study using 18 items adapted from Warner, Kerwin, and Walker’s [52] Sense of Community in Sport Scale (SCS), which assess six factors related to SOC: (1) administrative consideration (4 items, e.g., “I feel comfortable talking openly with the CrossFit coaches”), (2) common interest (3 items, e.g., “I share similar values with other members at CrossFit”), (3) equity in administrative decisions (2 items, e.g., “CrossFit coaches make decisions that benefit everyone”), (4) leadership opportunities (4 items, e.g., “I have influence over what CrossFit is like”), (5) social spaces (2 items, e.g., “When going to CrossFit, there are times when I can interact with other members”) and (6) competition (3 items, e.g., “I like the level of competition at CrossFit”). Each item was measured on a 4-point Likert scale (1 = not at all true; 2 = somewhat untrue; 3 = somewhat true; 4 = completely true), and was adapted to align specifically with CrossFit (i.e., using “CrossFit” and “coaches”, rather than general sporting terminology). A score for each of the six factors was calculated by averaging the appropriate items for each. Previous studies have established validity and reliability when the SCS is used [52], and similarly, our study sample resulted in an internal consistency of 0.90.

#### 2.2.6. Sociometric Network Data

Each CrossFit member was asked to report the names of all other members at their respective gyms they: (1) knew on a first name basis, (2) typically worked out with or saw at the gym, (3) would go to for advice or help with something, (4) would go to with a personal matter, and (5) spent time with outside of the gym. We combined results from the five prompts into one dichotomous network, where dyads that had any one (or more) of the five ties listed received a 1, and disconnected dyads received a 0.

### 2.3. Analytic Strategy

Descriptive statistics, including means, standard deviations, and frequencies were calculated for each gym and across the total sample using SPSS version 25 [53]. Linear network autocorrelation models (LNAMs) were computed to analyze the relationship between network connections and preference and tolerance, while accounting for demographic information, personality traits, depressive symptoms, and sense of community variables. By conducting LNAMs, we could assess whether CrossFit members’ preference and tolerance scores were associated with the scores of their social connections, while controlling for covariates. Network autocorrelation modeling is a form of regression analysis that uses permutation testing to specifically deal with the interdependent nature of network data. LNAMs determine the role network influences and connections may play in explaining specific outcome variables [54,55,56]. Six LNAMs were conducted predicting preference and tolerance for each gym, each computed using the statnet package [57] in R Studio version 4.0.0 [58]. Estimates were determined significant at the *p* < 0.05 level.

In addition to LNAMs, we used exponential random graph modeling (ERGMs) to determine significant factors associated with the presence of social connections between CrossFit members in each of the three gyms [59]. ERGMs use iterative Marcov Chain Monte Carlo algorithms to approximate the maximum likelihood estimates for the associations between a given set parameters (factors related to network structure or characteristics of the individuals in the network) and tie presence [59]. ERGMs return parameter estimates (PE) and standard errors (SE) for each factor entered into the model. These PEs are approximate log-odds representing increases or decreases in the probability of a tie existing between two people. A parameter is deemed significant at a *p* < 0.05 level if the PE is greater than two times the SE [55]. Three ERGMs were computed in this study, one for each gym. Cleaning and management of network data, along with LNAM and ERGM analyses, were completed using the statnet package [57] in R Studio [58].

#### Model Specification

We used the same set of variables in all six of the LNAMs, and the same set of parameters in all three ERGMs. Independent variables in the LNAMs included: gender (reference female), age, how long the respondent had been a member of CrossFit, personality traits (extraversion, agreeableness, conscientiousness, emotional stability, and openness to experiences), depressive symptoms scores, SOC variables (administrative consideration, common interest, equity in administrative decisions, leadership opportunities, social spaces, and competition), and average number of CrossFit classes the respondent attended per week.

For ERGMs, network structure parameters were added to model density (edges or connections in the network), reciprocity (connections that are mutually shared between two individuals), and transitivity (three individuals connected to each other). Further, parameters were added to understand the role of homophily, or similarity between people, related to gender, as well as preference and tolerance scores. Additionally, sender and receiver covariates were added for age, duration of CrossFit membership, average class attendance per week, depressive symptoms scores, personality traits, SOC variables, and preference and tolerance scores. These parameters assess the likelihood of incoming or outgoing ties with the increase (or decrease) in value for each variable. For example, we assessed whether higher depression scores were associated with a person sending connections, as well as a person receiving connections, within the three networks.

## 3. Results

### 3.1. Descriptive Statistics

Of the 197 participants across all three gyms, the mean age was 35.51 years (SD = 12.21, range = 19–77), the vast majority of the sample were white (87.8%, n = 173), and most respondents identified as female (65%, n = 128; please see Table 1 for sample characteristics for each gym, and across all respondents). Most participants (90.4%, n = 178) attended CrossFit classes three or more times per week, with an average of 4.02 classes per week (SD = 1.24). Just over 20% (21.3%, n = 42) of the sample had been members of CrossFit for a year or less, while 18.3% (n = 36) had been members of a CrossFit gym for four or more years. Average preference and tolerance scores for the whole sample were 13.64 (SD = 2.63, range = 7–20) and 13.74 (SD = 2.54, range = 8–20), respectively, with Gym 3 registering the highest average for preference, and Gym 2 registering the highest average for tolerance.

Personality scores were similar across all three gyms and ranged from 2–11 (out of a possible 14 for each dimension). This sample registered an average extraversion score of 6.55 (SD = 2.46), agreeableness score of 7.85 (SD = 1.89), conscientiousness score of 9.09 (SD = 1.34), emotional stability score of 7.65 (SD = 2.11), and openness to experiences score of 8.05 (SD = 1.51). The average PHQ-8 score was 4.07 (SD = 1.03, range = 0–24), with 5% (n = 10) of the sample registering scores above a 10, indicating evidence of severe depression. Gym 3 had the highest proportion of members who scored a 10 or higher on the PHQ-8.

Sense of community scores (out of possible score of 4 for each subscale) were also similar across all three gyms. Mean administrative consideration scores ranged from 2.18 to 3.25 across the total sample (M = 3.11, SD = 0.25), mean common interest scores were 2.78 (SD = 0.38, range = 1.22–3.11), decisions scores averaged at 2.72 (SD = 0.39, range = 1.50–3.00), mean leadership scores were 2.24 (SD = 0.60, range = 0.81–3.25), mean social spaces scores were 2.78 (SD = 0.35, range = 1.50–3.00), and competition scores averaged 2.65 (SD = 0.53, range = 0.78–3.11).

Gym 1 consisted of 58 people (nodes) with 1233 ties between them (edges). Gym 2 consisted of 31 people with 384 ties between them, and Gym 3 consisted of 98 people with 2736 ties between them. CrossFit members were connected to an average of 41.97 people (i.e., network degree, SD = 24.98) in Gym 1, 23.94 people (SD = 11.66) in Gym 2, and 49.94 people (SD = 37.88) in Gym 3. Networks ranged in density (the proportion of present ties to all potential ties within a network) from 0.23–0.41 (See Figure 1 for visuals of the gym networks), with Gym 3 representing the densest network. Table 1 outlines all descriptive statistics for each gym, as well as across the total sample.

### 3.2. Linear Network Autocorrelation Models

LNAMs from each gym explained between 18 and 55% of the variance in preference, and all three LNAMs revealed significant network effects on preference (Gym 1: β = 0.14, SE = 0.03; Gym 2: β = 0.21, SE = 0.05; Gym 3: β = 0.05, SE = 0.03). In addition to network effects, SOC variables were related to preference in all three gyms, including administrative consideration in Gym 1 (β = 3.97, SE = 1.97); common interests (β = 3.93, SE = 1.45), equity in administrative decisions (β = 5.28, SE = 1.21), leadership opportunities (β = 1.92, SE = 0.83) and social spaces (β = 2.79, SE = 1.91) in Gym 2; and competition (β = 2.44, SE = 0.71) in Gym 3. Demographic covariates, personality traits, and depressive symptoms were only related to preference in Gym 2. See Table 2 for all β and SE values from the three LNAMs conducted on preference, including covariates unique to each gym.

Regarding tolerance, LNAMs explained between 22 and 40% of variance, with network effects being associated with tolerance in all three gym networks (Gym 1: β = 0.19, SE = 0.02; Gym 2: β = 0.17, SE = 0.06; Gym 3: β = 0.07, SE = 0.02). In addition to network effects, SOC variables were related to tolerance in Gyms 1 and 2, with administration consideration (β = 4.14, SE = 1.60) related to tolerance in Gym 1, and administrative consideration (β = 0.79, SE = 0.27), equity in administrative decisions (β = 0.86, SE = 0.43), and social spaces (β = 0.96, SE = 0.40) related to tolerance in Gym 2. Further, personality traits were statistically significant in LNAMs from Gyms 2 and 3, with extraversion (β = 0.51, SE = 0.14), agreeableness (β = −0.47, SE = 0.21), conscientiousness (β = 0.68, SE = 0.29), and emotional stability (β = 0.50, SE = 0.20) related to tolerance in Gym 2, and conscientiousness (β = 0.60 SE = 0.20) related to tolerance in Gym 3. Demographic covariates and depressive symptoms were only related to tolerance scores in Gym 2. See Table 3 for all β and SE values from the LNAMs explaining tolerance, including unique factors for each network.

### 3.3. Exponential Random Graph Models

#### 3.3.1. Structural Properties

In all three gyms, structural properties including transitivity (having a friend in common; Gym 1: PE = 1.98, SE = 0.20; Gym 2: PE = 5.24, SE = 2.03; Gym 3: PE = 0.69, SE = 0.21) and reciprocity (Gym 1: PE = 6.90, SE = 0.69; Gym 2: PE = 2.34, SE = 0.27; Gym 3: PE = 2.45, SE = 0.08) increased the odds of a tie being present between people at CrossFit.

#### 3.3.2. Homophily

Gender homophily was related to the presence of social connections in Gym 2 (PE = 0.24, SE = 0.12) and Gym 3 (PE = 0.12, SE = 0.05), suggesting people of the same gender had higher odds of connecting in those networks. Homophily based on preference and tolerance was important in Gyms 1 and 3, although the direction of the relationship changed between Gym 1 and Gym 3. For Gym 1, having dissimilar preference scores (PE = −0.59, SE = 0.11), and similar tolerance scores (PE = 1.28, SE = 0.17) were associated with increased odds for a connection between people, whereas in Gym 3, those who had similar preference (PE = 0.03, SE = 0.01) and tolerance (PE = 0.06, SE = 0.01) scores were more likely to be connected.

#### 3.3.3. Non-Directional Covariates

The longer someone had been a CrossFit member, the more likely they were to connect with others within all three networks (Gym 1: PE = 1.74, SE = 0.17; Gym 2: PE = 0.16, SE = 0.05; Gym 3: PE = 0.20, SE = 0.01). Attending more classes per week was associated with increased odds of social connections in Gym 1 (PE = 1.84, SE = 0.07) and Gym 2 (PE = 0.16, SE = 0.06), and age was associated with social connections in Gyms 1 and 3, with older people more likely to make connections in Gym 1 (PE = 0.25, SE = 0.02) and younger people more likely to make connections in Gym 3 (PE = −0.01, SE = 0.00).

For Gyms 1 and 3, all five personality traits were related to the odds of connecting with others in CrossFit. Generally, higher conscientiousness scores decreased someone odds of making connections (Gym 1: PE = −0.24, SE = 0.12; Gym 3: PE = −0.12, SE = 0.01), while higher agreeableness (Gym 1: PE = 2.09, SE = 0.17; Gym 3: PE = 0.12, SE = 0.01), emotional stability (Gym 1: PE = 1.67, SE−0.14; Gym 3: PE = 0.07, SE = 0.01), and openness to experiences (Gym 1: PE = 1.42, SE = 0.17; Gym 3: PE = 0.06, SE = 0.01) scores were related to increased odds of making connections in both gyms. However, extraversion scores had different effects across Gym 1 and 3, with higher scores associated with less connections in Gym 1 (PE = −1.43, SE = 0.13), but more connections in Gym 3 (PE = 0.03, SE = 0.01).

While none of the SOC variables were related to ties in Gym 2, most were related to social connections within Gyms 1 and 3. The higher someone’s administrative consideration (Gym 1: PE = 6.71, SE = 0.31; Gym 3: PE = 0.16, SE = 0.02), equity in administrative decisions (Gym 1: PE = 1.64, SE = 0.46; Gym 3: PE = 0.10, SE = 0.03), leadership opportunities (Gym 1: PE = 2.48, S0.19; Gym 3: PE = 0.06, SE = 0.01), and social spaces (Gym 1: PE = 3.89, SE = 0.51; Gym 3: PE = 0.28, SE = 0.03) scores, the higher their odds of connecting with others members within Gyms 1 and 3.

#### 3.3.4. Sender/Receiver Covariates

While lower tolerance scores were associated with incoming ties in all gyms (Gym 1: PE = −0.31, SE = 0.10; Gym 2: PE = −0.11, SE = 0.05; Gym 3: PE = −0.03, SE = 0.01), higher preference scores were related to incoming ties in only Gyms 1 (PE = 0.23, SE = 0.07) and 2 (PE = 0.07, SE = 0.04). Depressive symptoms were associated with incoming and/or outgoing ties in all three networks, with slight variations across each gym. In Gym 1, greater depressive symptoms were related to someone sending outgoing connections (PE = 1.09, SE = 0.09), but negatively associated with incoming ties (PE = −0.99, SE = 0.09). In Gym 2, lower depressive symptoms (PE = −0.14, SE = 0.04) were related to incoming ties, while outgoing ties were showed no significant relationship. Finally, higher depressive symptoms scores (PE = 0.03, SE = 0.01) were related to outgoing ties in Gym 3, but depressive symptoms were not significantly related to incoming ties in the Gym 3 network. All parameter estimates and standard errors from ERGMs, including those unique to each network, can be found in Table 4.

## 4. Discussion

The purpose of this study was to explore the social environment of CrossFit relative to preference and tolerance scores among members. After conducting social network analyses on three gyms, we provide evidence that the preference and tolerance scores of individuals relate to the preference and tolerance scores of their social connections, suggesting the possibility of social influence. Further, we found preference and tolerance scores were important in explaining the social connections present within CrossFit gyms in varying ways.

### 4.1. Social Environment → Preference and Tolerance

This is the first study (to our knowledge) suggesting a relationship between the social environment (e.g., social connections, sense of community) and preference and tolerance scores among exercisers. While our findings are cross-sectional and would need follow-up data to provide more concrete evidence, we did find network effects related to preference and tolerance scores in all three gyms. These network effects could be due to social selection, social influence, and/or social context. In this case, social selection could result in people naturally connecting with CrossFit members who scored similarly on the PRETIE-Q. Contrarily, people’s preference and tolerance scores could be socially influenced by those of their peers, and therefore a person’s preference and tolerance scores adjust to mirror that of their social connections. Finally, as athletes are exposed to similar HIFT workouts, coaching, and social support within their social context, they could develop connections with others and build preference and tolerance for the workouts at the same time, resulting in similar preference and tolerance scores across ties within the network.

Previous literature defines and assumes preference and tolerance are fixed/stable traits [23,24,27], and therefore lend more support to the social selection hypothesis over social influence or social context, given the latter two assume the ability to develop preference and tolerance over time. However, the LNAMS in this study, which measures preference and tolerance as dependent variables, provides evidence for the possibility of social and contextual influence on preference and tolerance scores [54]. One explanation could be that even though preference and tolerance are dispositional characteristics, where certain people are predisposed to higher preference for and tolerance of high-intensity exercise than others, social and environmental factors create opportunities to express or manifest preference and tolerance differently than someone’s predisposition—sometimes referred to as *intraindividual variability* [60]. Research by William Fleeson [61] shows that the manifestation of personality traits, notably the Big Five (i.e., extraversion, agreeableness, conscientiousness, emotional stability, and openness to experiences), are contingent on situational factors, and that the intraindividual variability related to personality traits adjusts from situation to situation. For example, people might display more extraversion when connected to more extroverted people, or when immersed in a more social type situation. Though extraversion is a personality trait, the manifestation, or expression, of that trait becomes malleable given the situation [60,61,62]. Given most of the evidence that supports preference and tolerance as a stable trait has been conducted within an individual- versus group-exercise environment (e.g., running on a treadmill), it could be that someone in the CrossFit/group training environment (i.e., situation) might experience and express preference and tolerance differently than when exercising individually, particularly as someone connects with those who have higher preference and tolerance scores. These findings support other research that shows group-based exercise modalities, including yoga, bootcamp group workouts, and indoor cycling, can be more beneficial to a person’s health than individual/solo exercise due to the social environment available [17,63,64]. Longitudinal research is needed to further parse the impact and relative importance of social influence, social context, and social selection.

In addition to network effects, sense of community variables were related to preference and tolerance scores, providing further evidence for the role of the social environment in explaining preference and tolerance. Specifically, administrative consideration and common interests were two SOC factors related to preference and/or tolerance in two gyms. Administrative consideration refers to the care, support, and intentionality displayed from coaches to members within CrossFit [52]. Receiving this direct care and support, particularly in a HIFT-style workout, might help a person develop a stronger preference and tolerance for this type of exercise. Common interests describe commonality and belonging among CrossFit members. The relationship between common interests and preference and tolerance may support the selection hypothesis from above, in that, preference and tolerance may be a common value, or an interest shared between people, driving them to connect and relate.

The only demographic/background and personality trait variable related to preference and tolerance scores across multiple gyms was conscientiousness. Conscientiousness is defined as the tendency for an individual to follow socially prescribed norms for impulse control, to be goal-directed, to be planful, to delay gratification, and to follow norms and rules [65]. Conscientiousness has been linked to longevity [66], exercise behaviors [67], and overall better health [65], notably due to a person’s tendency toward achievement and order [66]. Research has also associated grit, or the perseverance and passion for long-term goals [68] with both conscientiousness and high-intensity exercise [69], which could be aspects of both preference and tolerance.

### 4.2. Preference and Tolerance → Social Connections

Exponential random graph models (ERGMs) using preference and tolerance scores as independent variables suggest preference and tolerance could be related to increased odds of social connections present within CrossFit networks. Specifically, people with similar tolerance scores (i.e., homophily based on tolerance) were more likely to connect in gyms 1 and 3. Additionally, tolerance scores were negatively related to incoming ties in all three gyms, meaning people with higher tolerance scores were less likely to be nominated in this study. These results support the social selection hypothesis as a likely reason for tolerance homophily, given the higher likelihood of two people with similar tolerance scores having a connection (i.e., selecting one another due to shared tolerance levels), while also having a lower likelihood of receiving many nominations, thus limiting their potential influence on others due to a smaller in-degree [70,71]. Qualitative studies indicate people have a tendency to be intimidated by CrossFit and high-intensity exercise, especially when they are newer to HIFT [15,19]. This could explain why those who demonstrate higher tolerance might receive less social connections—they could be intimidating due to their ability to tolerate higher intensity workouts.

While homophily based on tolerance was positively correlated with social ties across multiple gyms, the direction of the relationship between homophily based on preference and social ties varied from gym to gym. In Gym 1, having similar preference scores was negatively correlated with social connections, meaning people who scored similarly on the preference items of the PRETIE-Q were less likely to connect within this particular gym. Contrarily, in Gym 3, having similar preference scores was modestly related to a tie existing. Similarly, preference scores were positively related to outgoing ties in Gym 1 (i.e., people who had higher preference scores were more likely to nominate others in Gym 1), but negatively related to outgoing ties in Gym 3. Once again, the direction of the relationship between preference and social ties is inconsistent across gyms. Given the LNAMs reported earlier demonstrated consistent network effects related to preference scores when treated as the dependent variable, but the ERGMs were inconsistent in providing evidence of a relationship between preference as an independent variable and social connections, it could be more likely network effects related to preference are based on social and contextual influence, rather than selection [32], where social ties/the social environment prompt changes in the characteristic, rather than the characteristic prompting the creation of the social tie.

Overall, these findings suggest tolerance may have a more stable association with social connections across gyms, particularly as a possible driver behind social connections within CrossFit networks. Preference scores had more varied effects from network to network, and therefore may not be as prominent in driving social connections within CrossFit gyms. Additional research is needed to determine if preference and tolerance have similar associations with social connections as they did in this study.

In addition to preference and tolerance, structural variables, background/demographic variables, and SOC variables all helped explain the odds of social connections existing within multiple gyms. Structurally, reciprocity and transitivity were both associated with the presence of ties in all three gym networks. This suggests social connections created within CrossFit tend to be mutual (versus unidirectional), and members tend to have “friends in common” with their social ties, which often results in a highly clustered network with dense communities of people within the network. Reciprocity and transitivity both indicate a cohesive and strongly bonded network that could aid in long-term adherence to CrossFit [72,73]

Demographically, gender and age were related to the presence of ties in multiple gyms. Similar to previous literature, gender homophily was associated with social connections in gyms 2 and 3, meaning people of the same gender were more likely to connect with one another at CrossFit [74,75]. Like preference, age was also related to tie presence, but the direction of the relationship varied from gym to gym. In Gym 1, older participants were more likely to be connected with other members in their network, whereas in Gym 3, younger participants were more likely to be connected with others in their network, although the effect size is quite small in Gym 3. In Gym 1, it is possible that since the mean age is 25 years, some of the central people in the network (e.g., coaches, long-standing members), might be older, and therefore have more established connections in the network. Similarly, in Gym 3, where the mean age is slightly older (approximately 33 years), coaches and other central players may actually be younger than most members, driving the modest effect of age on social connections in this gym.

In all three gyms, having been a member of CrossFit for longer resulted in a larger likelihood of having social connections within the network. This makes sense, in that those who have been involved in the program longer have had an increased opportunity to make and sustain connections over time. Furthermore, those who have been members longer likely take leadership roles and are therefore connected with other members more readily based on their central position in the network [37,71]. The number of classes attended per week was also related to more social connections in two gyms (Gym 1 and 2), suggesting the more consistently someone attends classes, the more connected they are within their network. Thus, in addition to experiencing improved physical and mental health from more consistent class participation [76], members may also bolster social connections and social capital through regular class participation [21,77].

All five personality traits were related to the presence of ties in Gyms 1 and 3. Specifically, conscientiousness was negatively related to social ties; agreeableness, emotional stability, and openness to experiences were positively related to social ties; and extraversion positively related to social ties in Gym 3 and negatively related to Gym 1. This is unsurprising given personality largely influences the way people connect and interact with one another [46,61].

Finally, SOC variables including administrative consideration, equity in administrative decisions, leadership opportunities, and social spaces all positively related to social connections present within CrossFit networks. Sense of community broadly describes the ways a person feels belonging and support as a member of a larger group [51], and therefore should be associated with someone’s connections they make within that network. These findings provide a rationale for CrossFit coaches and administrators to prioritize building a sense of community for their members, notably through administrative consideration, equity of administrative decisions, leadership opportunities, and social spaces. These efforts could result in more social connectivity among members, which often result in longer-term adherence, better physical and mental health benefits, and overall a greater enjoyment in the program [9,19,20].

### 4.3. Limitations

While this is the first study to suggest the possibility of social influence on preference for and tolerance of high-intensity exercise, the cross-sectional design limits our ability to draw concrete conclusions. Similarly, despite inviting all members from each gym to participate, those who did choose to participate could bias the findings of our study. Each network we measured had missing data, and while we were confident our response rate was high enough to draw accurate conclusions from the represented network [41,42], it is possible that key members of the network, or important social ties, were left out. Similarly, people who might have joined CrossFit but left prior to data collection may not compare to those who remained members. These people who left might not benefit from the social environment the same ways as our sample participants did. Finally, the psychometric properties of the PRETIE-Q did not lend itself to retaining the original 16 items in our sample, and the TIPI only presented a moderate internal consistency. Future research may wish to investigate a shortened and revised version of this measure for use in varied exercise contexts. Additionally, future research employing longitudinal and/or experimental designs would better determine if preference and tolerance really do manifest differently in group settings as compared to independent exercise situations.

## 5. Conclusions

Using social network analyses, we were able to demonstrate evidence for the social influence of preference and tolerance within CrossFit networks. Because preference and tolerance are important factors related to a person enjoying and adhering to high-intensity exercise, understanding situations and environments where preference and tolerance could be optimized might result in increased levels of beneficial exercise among adults. This study also supported previous research highlighting the importance of the social environment created within CrossFit, and was the first to use social network analysis to investigate what drives social connections within the CrossFit environment.

## Figures and Tables

**Figure 1 ijerph-17-08370-f001:**
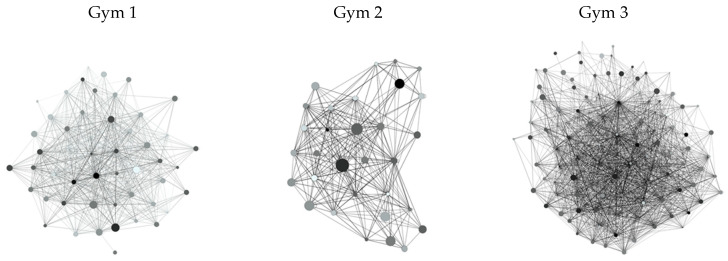
Visualizations of gym networks. Nodes vary in size based on preference score (larger nodes indicate higher preference scores), and nodes are shaded based on tolerance scores (darker nodes indicate higher tolerance scores).

**Table 1 ijerph-17-08370-t001:** Sample characteristics of members from three CrossFit gyms.

	Gym 1		Gym 2		Gym 3		Total Sample	
	M ± SD	%, n	M ± SD	%, n	M ± SD	%, n	M ± SD	%, n
Age	25.74 ± 10.22		43.58 ± 19.05		33.07 ± 9.52		35.51 ± 12.21	
Gender								
Female		58.6%, n = 34		58.1%, n = 18		70.4%, n = 76		65.0%, n = 128
Male		41.4%, n = 24		41.9%, n = 13		29.6%, n = 32		35.0%, n = 69
Race								
White		81%, n = 47		87.1%, n = 27		91.7%, n = 99		87.8%, n = 173
Black		0%, n = 0		0%, n = 0		0%, n = 0		0%, n = 0
Hispanic		1.7%, n = 1		0%, n = 0		0%, n = 0		0.5%, n = 1
Asian		1.7%, n = 1		0%, n = 0		0%, n = 0		0.5%, n = 1
Native American		1.7%, n = 1		0%, n = 0		0%, n = 0		0.5%, n = 1
Multiracial		3.4%, n = 2		3.2%, n = 1		0%, n = 0		1.5%, n = 3
Other		6.9%, n = 4		3.2%, n = 1		2.8%, n = 3		4.1%, n = 8
Prefer not to say		3.4%, n = 2		0%, n = 0		0%, n = 0		1.0%, n = 2
Classes per Week	4.48 ± 1.14		3.55 ± 1.18		3.91 ± 1.23		4.02 ± 1.24	
<1/week		1.7%, n = 1		0%, n = 0		0.9%, n = 1		1.0%, n = 2
1/week		0%, n = 0		0%, n = 0		4.6%, n = 5		2.5%, n = 5
2/week		3.4%, n = 2		22.6%, n = 7		2.8%, n = 3		6.1%, n = 12
3/week		8.6%, n = 5		29.0%, n = 9		25.0%, n = 27		20.8%, n = 41
4/week		31.0%, n = 18		22.6%, n = 7		38.0%, n = 41		33.5%, n = 66
5/week		39.7%, n = 23		22.6%, n = 7		18.5%, n = 20		25.4%, n = 50
>5/week		15.5%, n = 9		3.2%, n = 1		10.2%, n = 11		10.7%, n = 21
CrossFit Member								
<6 months		19%, n = 11		3.2%, n = 1		9.3%, n = 10		11.2%, n = 22
6 months–1 year		5.2%, n = 3		22.6%, n = 7		9.3%, n = 10		10.2%, n = 20
>1–2 years		12.1%, n = 7		9.7%, n = 3		23.1%, n = 25		17.8%, n = 35
>2–3 years		29.3%, n = 3		12.9%, n = 4		14.8%, n = 16		18.8%, n = 37
>3–4 years		20.7%, n = 12		29.0%, n = 9		24.1%, n = 26		23.9%, n = 47
4+ years		13.8%, n = 8		22.6%, n = 7		19.4%, n = 21		18.3%, n = 36
Preference	13.30 ± 2.67		12.40 ± 2.84		14.17 ± 2.42		13.64 ± 2.63	
Tolerance	13.78 ± 2.12		14.19 ± 2.46		13.60 ± 2.77		13.74 ± 2.54	
Depressive Symptoms	4.19 ± 2.89		3.06 ± 2.83		4.29 ± 4.01		4.07 ± 3.55	
Severe (+10)		1.7%, n = 1		0%, n = 0		8.4%, n = 9		5.0%, n = 10
Personality								
Extraversion	6.57 ± 2.61		6.47 ± 2.18		6.57 ± 2.48		6.55 ± 2.46	
Agreeableness	7.93 ± 1.29		8.15 ± 1.97		7.72 ± 1.91		7.85 ± 1.89	
Conscientiousness	9.03 ± 1.65		9.29 ± 1.14		9.06 ± 1.22		9.09 ± 1.34	
Emotional Stability	7.64 ± 2.16		7.63 ± 2.11		7.65 ± 2.10		7.65 ± 2.11	
Openness to Experiences	7.83 ± 1.37		8.45 ± 1.39		8.06 ± 1.59		8.05 ± 1.51	
Sense of Community								
Administrative	3.13 ± 0.21		3.03 ± 0.33		3.11 ± 0.26		3.11 ± 0.25	
Consideration								
Common Interest	2.74 ± 0.36		2.71 ± 0.46		2.84 ± 0.36		2.78 ± 0.38	
Equity in Administrative	2.76 ± 0.30		2.57 ± 0.39		2.74 ± 0.42		2.72 ± 0.39	
Decisions								
Leadership Opportunities	2.16 ± 0.53		2.42 ± 0.66		2.24 ± 0.61		2.24 ± 0.60	
Social Spaces	2.85 ± 0.28		2.63 ± 0.46		2.79 ± 0.33		2.78 ± 0.35	
Competition	2.70 ± 0.43		2.41 ± 0.73		2.70 ± 0.49		2.65 ± 0.53	
Network Descriptives								
Network Nodes, Edges	58, 1233		31, 384		98, 2736			
Network Degree	41.97 ± 24.98		23.94 ± 11.66		49.94 ± 37.88			
Network Density	0.37		0.41		0.23			

**Table 2 ijerph-17-08370-t002:** Linear network autocorrelation models assessing preference scores in three CrossFit networks.

Covariate	Gym 1: R^2^ = 0.35 ***		Gym 2: R^2^ = 0.55 ***		Gym 3: R^2^ = 0.18 ***	
	β	SE	β	SE	β	SE
Gender (ref: female)	0.28	0.82	−3.04 ***	0.64	−0.35	0.63
Age	−0.05	0.23	0.06 *	0.03	0.50	0.03
Classes per Week	0.86 *	0.40	0.69 *	0.33	−0.57	0.28
CrossFit Member	0.08	0.27	0.14	0.41	0.58	0.19
Personality						
Extraversion	0.03	0.15	−0.32	0.21	1.46	0.12
Agreeableness	−0.02	0.21	−0.23	0.33	0.45	0.17
Conscientiousness	0.29	0.22	0.44	0.44	1.22	0.25
Emotional Stability	0.16	0.18	0.15	0.28	−0.29	0.16
Openness to Experiences	0.40	0.31	0.51	0.28	−0.21	0.20
Depressive Symptoms	0.25	0.13	−0.54 **	0.19	0.16	0.08
Sense of Community						
Administrative Consideration	3.97 **	1.97	0.92	1.66	0.48	1.31
Common Interest	−1.83	1.29	3.93 **	1.45	0.82	1.12
Equity in Administrative Decisions	2.09	1.460	5.28 ***	1.21	0.93	0.94
Leadership Opportunities	−1.23	0.92	1.92 *	0.83	−1.03	0.56
Social Spaces	−0.08	1.76	2.79	1.91	−0.32	1.23
Competition	1.21	1.14	0.26	0.81	2.44 *	0.71
Network Effects	0.14 ***	0.03	0.21 ***	0.05	0.05 **	0.03

Note. * *p* < 0.05, ** *p* < 0.01, *** *p* < 0.001.

**Table 3 ijerph-17-08370-t003:** Linear network autocorrelation models assessing tolerance scores in three CrossFit networks.

Covariate	Gym 1: R^2^ = 0.22 ***		Gym 2: R^2^ = 0.40 ***		Gym 3: R^2^ = 0.23 ***	
	β	SE	β	SE	β	SE
Gender (ref: female)	0.10	0.67	−1.24 ***	0.45	−0.37	0.50
Age	0.00	0.03	0.06	0.28	0.01	0.03
Classes per Week	0.47	0.33	0.41 *	0.14	0.10	0.22
CrossFit Member	0.08	0.19	0.06	0.23	0.14	0.10
Personality						
Extraversion	0.08	0.18	0.51 ***	0.14	−0.04	0.10
Agreeableness	0.08	0.13	−0.47 *	0.21	−0.09	0.14
Conscientiousness	0.01	0.19	0.68 *	0.29	0.55 **	0.20
Emotional Stability	0.09	0.15	0.50 *	0.20	0.22	0.12
Openness to Experiences	0.09	0.25	0.13	0.19	0.10	0.16
Depression	−0.02	0.11	−0.35 **	0.11	0.12	0.06
Sense of Community						
Administrative Consideration	1.13 *	0.50	0.79 **	0.27	0.38	0.27
Common Interest	−0.21	0.35	0.58	0.32	−0.02	0.31
Equity in Administrative Decisions	0.23	0.63	0.86 *	0.43	−0.19	0.38
Leadership Opportunities	0.08	0.30	−0.12	0.15	−0.03	0.11
Social Spaces	0.63	0.71	0.96 *	0.40	0.06	0.49
Competition	−0.26	0.34	−0.24	0.18	0.31	0.20
Network Effects	0.19 **	0.02	0.17 ***	0.06	0.07 **	0.02

Note. * *p* < 0.05, ** *p* < 0.01, *** *p* < 0.001.

**Table 4 ijerph-17-08370-t004:** Exponential random graph models assessing factors related to tie presence in three CrossFit networks.

Parameter	Gym 1		Gym 2		Gym 3	
	Estimate	SE	Estimate	SE	Estimate	SE
Structural						
Edges	12.70 ***	1.16	13.07 ***	3.31	3.08 ***	0.63
Reciprocity	6.90 ***	0.69	2.34 ***	0.27	2.45 ***	0.08
Transitivity	1.98 **	0.20	5.24 **	2.03	0.69 ***	0.21
Homophily						
Gender	−0.40	0.45	0.24 *	0.12	0.12 **	0.05
Preference	−0.59 ***	0.11	−0.01	0.03	0.03 *	0.01
Tolerance	1.28 ***	0.17	0.06	0.04	0.06 ***	0.01
Non-Directional Covariates						
Age	0.25 ***	0.02	−0.00	0.00	−0.01 ***	0.00
CrossFit Member	1.74 ***	0.17	0.16 ***	0.05	0.20 ***	0.01
Classes per Week	1.84 ***	0.19	0.16 **	0.06	−0.01	0.02
Extraversion	−1.43 ***	0.13	−0.03	0.04	0.02 *	0.01
Conscientiousness	−0.24 *	0.12	0.07	0.06	−0.12 ***	0.01
Agreeableness	2.09 ***	0.15	0.02	0.05	0.12 ***	0.01
Emotional Stability	1.67 ***	0.14	−0.02	0.04	0.07 ***	0.01
>Openness to Experiences	1.42 ***	0.17	−0.02	0.04	0.06 ***	0.01
Administrative Consideration	6.71 ***	0.31	0.02	0.07	0.16 ***	0.02
Common Interest	0.38	0.21	−0.01	0.07	0.15 ***	0.02
Equity in Administrative Decisions	1.64 ***	0.46	0.07	0.11	0.10 ***	0.03
Leadership Opportunities	2.48 ***	0.19	−0.02	0.03	0.06 ***	0.01
Social Spaces	3.89 ***	0.51	−0.08	0.08	0.28 ***	0.03
Competition	−2.68 ***	0.29	0.05	0.04	−0.02	0.01
Sender/Receiver Covariates						
Depressive Symptoms (In)	−0.99 ***	0.09	−0.14 **	0.04	0.00	0.01
Depressive Symptoms (Out)	1.09 ***	0.09	0.03	0.04	0.03 ***	0.01
Preference (In)	0.23 ***	0.07	0.07 *	0.04	−0.00	0.01
Preference (Out)	0.67 **	0.07	−0.05	0.04	−0.03 **	0.01
Tolerance (In)	−0.31 **	0.10	−0.11 *	0.05	−0.03 *	0.01
Tolerance (Out)	−0.15	0.10	0.11 *	0.05	−0.00	0.01

Note. * *p* < 0.05, ** *p* < 0.01, *** *p* < 0.001.

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
