# Peer review of "Network Analysis of the Social Environment Relative to Preference for and Tolerance of Exercise Intensity in CrossFit Gyms"

_ijerph, 2020, doi:10.3390/ijerph17228370_

Round 1

Reviewer 1 Report

You all have done a great job with the research study.  It was very interesting to read and is very applicable.  It is obvious you have taken the time to write and review the article before submitting.

I have minimal feedback and questions:

Do you have demographic information for the gyms as a whole, not just those who completed the study? This would be beneficial to include and interesting to see what members from each gym completed the study.  Does this add any type of bias?

In the discussion, can you compare your findings to other types of training and/or types of gyms? How does a CrossFit gym compare to other locations or training types?

A potential limitation is only assessing those who are currently members of a CF gym.  Assessing those who have left the gym may provide valuable information.

It is worth noting if all participants were participating in the prescribed workout each day they attended, or something else.  State if any of the gyms provided various workouts for various levels of fitness or skill levels.

Typically CrossFit gyms play music during each workout.  Could this play a role in your results and the tolerance for HIFT?

Again, great job with this study! 

Reviewer 2 Report

First of all, I appreciate that you have sent me this manuscript for review. The subject is interesting, but there are some minor considerations in the study that authors should review:

Title:

It comprises all the key concepts of the paper

Abstract:

Its content fits the right structure

Keywords:

They could be more representative of the paper if (at least partially) they refers to the measured variables (e.g. “deppresive symptoms”, “sense of community”, “personality”…)

Introduction:

World Health Organization’s guidelines for exercise could be mentioned.

In line 64, write “and” instead of “&”.

Measures:

In section 2.2.1, author/s should indicate the data collection of skin color, as reported in line 251 and Table 1.

In section 2.2.2. it would be interesting to include explicitly one ítem (in terms of example) of the two final factors.

In section 2.2.3., line 157, review the bibliographic rules of this journal.

In section 2.2.4, lines 176 and 184, review the bibliographic rules of this journal.

In section 2.2.5., indicate how many ítems each factor has.

Results:

They are described enough and appropriate.

Discussion:

In my view, this is one of the most important section of a paper. I congratulate the author/s for this current state.

Conclusions:

Its content fits the right structure

References:

According to IJERPH guidelines (available at their webpage), “References should be described as follows, depending on the type of work:

  • Journal Articles:
    1. Author 1, A.B.; Author 2, C.D. Title of the article. Abbreviated Journal NameYearVolume, page range.”

This requirement is not met many times.

When possible, try to update references, including new references from 2019 and 2020.

Reviewer 3 Report

My comments are predominantly minor for the content, but do have some very minor edits/suggestions to improve readability:

1) Title: The title is correct as it reflects correctly the objective and hypothesis of the work.

2) Summary: This section follow a well structured format.

3) Introduction: The research question itself is sound and the topic is strongly introduced. On the other hand, Introduction section may be improved adding new information in order to provide an adequate state-of-the-art including some references. I suggest to include this references include in the atteched to complet this requeriment  relative to thermography and soccer injuries

Rodriguez-Sanz, D.; Losa-Iglesias, M.E.; Becerro de Bengoa-Vallejo, R.; Palomo-Lopez, P.; Beltran-Alacreu, H.; Calvo-Lobo, C.; Navarro-Flores, E.; Lopez-Lopez, D. Skin temperature in youth soccer players with functional equinus and non-equinus condition after running. J. Eur. Acad. Dermatology Venereol. 2018, 32, 2020–2024.

4) Materials and Methods: The inclusion and exclusion criteria are not adequate. Sampling bias needs to be discussed within the limitations section given the study design. Participants are not reported to have consented appropriately and the protocol was approved by an ethics committee.

Moreover I suggest  authors must include a reference to Ethics requirements Helsinki declaration and Strobe methods

-Vandenbroucke, J.P.; von Elm, E.; Altman, D.G.; Gøtzsche, P.C.; Mulrow, C.D.; Pocock, S.J.; Poole, C.; Schlesselman, J.J.; Egger, M.; STROBE Initiative Strengthening the Reporting of Observational Studies in Epidemiology (STROBE): explanation and elaboration. Int. J. Surg. 201412, 1500–24.

-Holt, G.R. Declaration of Helsinki—The World’s Document of Conscience and Responsibility. South. Med. J. 2014107, 407–407.

5) Results: The results is clear and concise with appropriate statistical analysis been performed appropriately and rigorously.

6) Discussion: The discussion appears well developed and appropriate, authors describe the results, the limitations and compare with other researchs. The limitations section needs to incorporate the sampling bias given the study design.

7) Conclusion: The conclusion is conclusively.

8) References: Appropriate

9) Figures and tables: Correct

Reviewer 4 Report

Dear author,

Thank you for your considerable work. The manuscript presents high scientific merit and the information showed the importance of the social environment created within CrossFit and the social influence of preference and tolerance within CrossFit networks, is quite interesting to understand. However, some minor issues can be addressed to improve the quality and comprehensibility:

METHODS SECTION

  1. In Ln 110-11, please add the specific name/names of the institutional review board.
  2. In Ln 123-128, please check the spaces between numbers and symbols (i.e., male = 0 or 1=6 months). Please, check this issue along the results section as well.
  3. In Ln 147, please change PRETIEQ with PRETIE-Q.
  4. In Ln 208, please remove the comma after “frequencies”.
  5. In Ln 209, please add the characteristics of SPPS version 25 used (i.e., SPSS, V.25.0, IBM SPSS Statistics, IBM Corporation). Please, do the same for R studio in Ln218.
  6. The internal consistency of the TIPI in your study is lower than the “acceptable” Cronbach’s α. I should be included as a limitation in the discussion section.
  7. In Ln 219, please include the abbreviation of exponential random graph models here (this is the first time you mentioned it in the text).
  8. The authors did not specify the significance value for the LNAMs, please add it.

DISCUSSION SECTION

  1. In Ln 375-382, the reviewer suggests rewriting those sentences trying to make them shorter, and avoiding the definition of concepts such as social selection or influence. If the authors consider that information as essential, they should include those definitions in the introduction section.
  2. In Ln 425-427, please rewrite those sentences and do not repeat the information showed in the results section.
  3. I think that the authors should add a little discussion about the results found with depressive symptoms due to the health benefits of exercise.
